# *Pseudomonas aeruginosa*-Derived DnaJ Induces the Expression of IL−1β by Engaging the Interplay of p38 and ERK Signaling Pathways in Macrophages

**DOI:** 10.3390/ijms242115957

**Published:** 2023-11-03

**Authors:** Dae-Kyum Kim, Jin-Won Huh, Hyeonseung Yu, Yeji Lee, Yongxin Jin, Un-Hwan Ha

**Affiliations:** 1Department of Biotechnology and Bioinformatics, Korea University, Sejong 30019, Republic of Korea; eorua1110@korea.ac.kr (D.-K.K.); huh1056@korea.ac.kr (J.-W.H.); cockychild74@naver.com (H.Y.); yejee90@korea.ac.kr (Y.L.); 2Interdisciplinary Graduate Program for Artificial Intelligence Smart Convergence Technology, Korea University, Sejong 30019, Republic of Korea; 3State Key Laboratory of Medicinal Chemical Biology, Key Laboratory of Molecular Microbiology and Technology of the Ministry of Education, Department of Microbiology, Nankai University, Tianjin 300071, China; yxjin@nankai.edu.cn

**Keywords:** HSP40 homolog, IL−1β, inflammasome, MAPK, NF-κB

## Abstract

As members of pathogen-associated molecular patterns, bacterial heat shock proteins (HSPs) are widely recognized for their role in initiating innate immune responses. This study aimed to examine the impact of DnaJ, a homolog of HSP40 derived from *Pseudomonas aeruginosa* (*P. aeruginosa*), on the regulation of IL−1β expression in macrophages. We demonstrated that DnaJ modulates macrophages to secrete IL−1β by activating NF-κB and MAPK signaling pathways. Specifically, ERK was identified as a positive mediator for IL−1β expression, while p38 acted as a negative mediator. These results suggest that the reciprocal actions of these two crucial MAPKs play a vital role in controlling IL−1β expression. Additionally, the reciprocal actions of MAPKs were found to regulate the activation of inflammasome-related molecules, including vimentin, NLRP3, caspase-1, and GSDMD. Furthermore, our investigation explored the involvement of CD91/CD40 in ERK signaling-mediated IL−1β production from DnaJ-treated macrophages. These findings emphasize the importance of understanding the signaling mechanisms underlying IL−1β induction and suggest the potential utility of DnaJ as an adjuvant for stimulating inflammasome activation.

## 1. Introduction

*Pseudomonas aeruginosa* (*P. aeruginosa*) is a well-known bacterial pathogen, possessing a wide range of virulence factors that act as pathogen-associated molecular patterns (PAMPs), triggering innate defense responses. It is an opportunistic pathogen with the ability to cause a diverse range of severe and potentially life-threatening infections, especially in individuals with compromised immune systems, such as patients and the elderly [1]. Moreover, *P. aeruginosa* is recognized as a significant contributor to infections due to its notable resistance to antibiotics [2]. Recognition of the PAMPs occurs through the action of pattern recognition receptors (PRRs), leading to the production of various proinflammatory cytokines, including interleukin-1β (IL−1β), which plays a crucial role in stimulating early immune responses [3]. IL−1β plays a vital role in host protection against infections and injuries [4], and it is induced by various pathogens, including bacteria and viruses [5,6,7,8]. The production of IL−1β involves two distinct steps, known as priming and activation, in various cell types, including macrophages [9]. Initially, activated nuclear factor-κB (NF-κB) triggers the production of pro-IL−1β, followed by inflammasome signaling activation, which induces the posttranslational cleavage of 31 kDa pro-IL−1β into its active form through the action of enzymatically active caspase-1 [10,11]. The active caspase-1 is a heterodimer composed of subunits p10 and p20, which are generated by the cleavage of 45 kDa pro-caspase-1 [12]. In addition to cleaving pro-IL−1β, the caspase-1 specifically recognizes and cleaves gasdermin D (GSDMD), which facilitates the formation of pores in the plasma membrane, resulting in pyroptosis, a unique form of programmed cell death [13].

Innate defense responses play crucial roles in effectively eliminating invading pathogens, and extensive research has focused on the role of IL−1β-mediated acute inflammatory responses in microbial infections, including those caused by *P. aeruginosa*, *Staphylococcus aureus*, and influenza [14,15,16]. These initial host defense mechanisms are often initiated by the recognition of microbial products released during infection. Among these microbial products, heat shock proteins (HSPs) are a highly conserved group of proteins found ubiquitously in both prokaryotic and eukaryotic cells. They are synthesized in response to various environmental stresses and function as molecular chaperones, aiding in the proper folding and unfolding of proteins [17]. Interestingly, bacterial HSPs serve as PAMPs capable of strongly triggering inflammatory immune responses in macrophages [18]. Expanding upon this knowledge, we have previously reported that DnaK, a homolog of HSP70, regulates the production of IL−1β by modulating the activation of NF-κB/JNK and PI3K/FoxO1 signaling pathways [19,20]. In addition, it was shown that HtpG, an HSP90 homolog in *P. aeruginosa*, regulates the expression of IL−8 through TLR4 and CD91 [21].

In the context of DnaK’s chaperone activity, DnaJ plays a crucial role as an essential co-chaperone, stimulating the ATPase and protein-folding activities of DnaK. Notably, DnaJ, recognized as a virulence factor, holds importance in the pathogenesis of bacterial infections. Various studies have demonstrated the indispensable nature of the DnaK/DnaJ chaperone system in *Salmonella enterica* for bacterial survival within macrophages and the invasion of epithelial cells [22]. DnaJ is also essential for biofilm formation and pyocyanin production in *P. aeruginosa* [23], and DnaJ derived from *Streptococcus pneumoniae* (*S. pneumoniae*) plays a role in promoting Th1 and Th17 immune responses by activating BMDC and stimulating the proliferation of CD4^+^ T cells in a TLR4-dependent manner [24]. Moreover, *S. pneumoniae*-derived DnaJ elicits a robust immune response [25], highlighting its potential as a PAMP capable of triggering host immune responses, and recently, we reported the role of DnaJ in inducing the expression IL−8 [26]. However, the specific roles of *P. aeruginosa*-derived DnaJ in inducing inflammatory responses and the underlying molecular mechanisms still require further investigation.

In this study, we aimed to explore the potential roles of DnaJ in modulating inflammasome activation and IL−1β production, providing novel insights into immune response regulation. Our findings uncover the involvement of the extracellular regulated protein kinases (ERK) and p38 mitogen-activated protein kinases (MAPKs) signaling pathways in the regulation of IL−1β expression by controlling inflammasome activation in macrophages. Additionally, we investigated the potential involvement of CD91/CD40 in the ERK signaling-mediated activity of IL−1β production in DnaJ-treated macrophages. These discoveries establish DnaJ as a novel regulator of IL−1β production through the activation of the inflammasome.

## 2. Results

### 2.1. P. aeruginosa-Derived DnaJ Induces the Expression of IL−1β

In our previous study, we demonstrated the upregulation of IL−1β expression in response to DnaK derived from *P. aeruginosa* [19,20]. To assess the impact of *P. aeruginosa*-derived DnaJ, acting as a co-chaperone of DnaK, we obtained the protein using the previously described method [27]. It is worth mentioning that we employed *Escherichia coli* containing an empty vector to generate a control extract. The control extract was produced using the same purification procedure and served as a comparative control in our experimental setup. As shown in Figure 1A, we examined the involvement of the DnaJ protein in this induction by treating the samples with proteinase K, a commonly used serine protease for protein digestion in biological samples. Notably, the induction was almost completely abolished upon treatment with proteinase K, indicating that DnaJ is a potent stimulator of IL−1β expression. Additionally, we observed a dose- and time-dependent increase in both mRNA and protein levels, with the maximum induction reached after 4 h of treatment (Figure 1B–D). Taken together, these findings strongly support the notion that DnaJ derived from *P. aeruginosa* possesses the capability to induce IL−1β production.

### 2.2. Reciprocal Effects of p38 and ERK MAPK Signaling Pathways for the Expression of IL−1β

The NF-κB and MAPK signaling pathways are well-known for their involvement in the regulation of IL−1β expression [28,29]. Given the induction of IL−1β in response to DnaJ treatment, our objective was to investigate the roles of these signaling mediators. To examine this, cells were pretreated with specific chemical inhibitors prior to DnaJ treatment. As shown in Figure 2A,B, both the mRNA and protein levels of IL−1β significantly decreased after pretreatment with BAY11-7082, an NF-κB chemical inhibitor, and PD98059, an ERK chemical inhibitor. Conversely, the levels significantly increased following pretreatment with SB203580, a p38 chemical inhibitor, suggesting a negative involvement of p38 in IL−1β expression, while NF-κB and ERK displayed positive involvement. Considering that ERK consists of two isoforms, ERK1 and ERK2 [30], we conducted an siRNA interference assay to determine which isoform is responsible for the regulation. As shown in Figure 2C, silencing ERK1 reduced the expression of IL−1β, whereas ERK2 silencing did not clearly impede the expression, indicating the involvement of ERK1 in the regulation of IL−1β expression in response to DnaJ treatment. The effects of siERK1 and siERK2 were validated by measuring the mRNA levels of ERK1 and ERK2 (Figure 2D,E). To assess p38 and ERK activation, we evaluated the phosphorylation levels in response to DnaJ treatment (Figure 2F), and the analysis indicated that phosphorylation levels of p38 and ERK reached their peaks at 30 min and 60 min, respectively.

We observed that DnaJ treatment led to a time-dependent enhancement in the production of mature IL−1β (Figure 1D). Additionally, this treatment influenced production in a MAPK-dependent manner (Figure 2B), indicating the differential regulation of inflammasome activation. To examine the effects of p38 and ERK on mediators associated with inflammasome signaling, cells were pretreated with inhibitors of p38 and ERK prior to DnaJ treatment. As shown in Figure 2G, DnaJ treatment resulted in an increase in the expression of vimentin, NLRP3, the cleaved form of caspase-1 p20, and cleavage of GSDMD. Interestingly, inhibition of p38 led to an elevation in the expression of these markers, suggesting a negative involvement of p38 in their expression. However, inhibition of ERK had minimal effect. Considering that NF-κB acts as a positive regulator for expression (Figure 2A), we examined the cross-talk between NF-κB and MAPKs by immunoblotting analysis in the presence of specific inhibitors. As shown in Figure 2H,I, inhibition of NF-κB and MAPKs had little impact on the activation of MAPKs, as indicated by p38 and ERK phosphorylation, as well as the activation of NF-κB, as indicated by IKKαβ phosphorylation. Thus, the production of IL−1β mediated by DnaJ is regulated by the p38 and ERK signaling pathways, which function as distinct signals to activate inflammasome signaling.

### 2.3. DnaJ-Induced IL−1β Expression Is under the Control of CD91/CD40 Signaling Pathway

In the context of the innate immune response, HSPs interact with signaling receptors to initiate the release of inflammatory cytokines by macrophages [31]. CD91 and CD40 have been identified as a common receptor for HSP70 [32,33]. To investigate the involvement of the CD91 and CD40, we conducted an siRNA interference assay using CD91 siRNA (siCD91) and CD40 siRNA (siCD40) in dTHP-1 macrophage-like cells. As shown in Figure 3A,B, the interference partially reduced the expression of IL−1β in a dose-dependent manner. Co-transfection of both siRNAs further decreased the levels of mRNA and protein production (Figure 3C,D), indicating that the induction is mediated by both signaling pathways. The effects of siCD91 and siCD40 were validated by measuring the mRNA levels of each target (Figure 3E,F). Next, we examined the effect of CD91 and CD40 on the activation of p38, ERK, and NF-κB using siRNA interference. Immunoblotting analysis revealed that the phosphorylation of p38 and IKKαβ remained unchanged, while the phosphorylation of ERK was decreased upon siRNA-mediated suppression of CD91 and CD40 (Figure 3G). Furthermore, co-interference of both siRNAs significantly decreased the phosphorylation of ERK, indicating that these receptors contribute to the control of ERK signaling. Therefore, this study indicates that CD91/CD40 and DnaJ may be primarily associated with the induction of IL−1β expression via the ERK signaling pathway in dTHP-1 cells.

### 2.4. DnaJ-Induced IL−1β Expression Is Partially under the Control of TLR2/TLR4 Signaling Pathway

Toll-like receptors (TLRs), particularly TLR2 and TLR4, play crucial roles in the host’s innate recognition system against bacterial components [34]. To investigate the involvement of these TLRs, we pretreated dTHP-1 cells with OxPAPC, a chemical inhibitor of TLR2 and TLR4, to assess their impact. As shown in Figure 4A, the pretreatment dose-dependently decreased the expression of IL−1β, indicating the involvement of TLR2/4 signaling in its induction. To determine the specific contribution of each TLR, we initially used CLI-095, a specific chemical inhibitor of TLR4, and observed a partial reduction in IL−1β expression at the mRNA level in a dose-dependent manner (Figure 4B). Subsequently, we examined whether the DnaJ-mediated activation of p38, ERK, and NF-κB is controlled by TLR4 by measuring their phosphorylation levels in the presence and absence of CLI-095. As shown in Figure 4C, pretreatment with the inhibitor did not lead to any differences in phosphorylation, suggesting that these signaling pathways are not regulated by the TLR4 signaling pathway. Next, we investigated the effect of TLR2 on the DnaJ-mediated expression of IL−1β by using siRNA interference. As shown in Figure 4D, transfection with TLR2 siRNA (siTLR2) resulted in a partial decrease in expression. The knockdown of TLR2 by the siRNA was confirmed through immunoblotting analysis. Subsequently, we examined whether the DnaJ-mediated activation of p38, ERK, and NF-κB is controlled by TLR2 by measuring their phosphorylation levels following interference. As shown in Figure 4E, the interference did not induce any differences in phosphorylation, suggesting that these signaling pathways are not regulated by the TLR2 signaling pathway. Taken together, these observations indicate that DnaJ-mediated IL−1β expression is regulated in a TLR2/4-dependent manner. However, the activation of p38 and NF-κB is not controlled by these receptors.

## 3. Discussion

In response to various microbial products, inflammasomes are activated through two distinct steps: priming and activation. This process triggers the production of biologically active IL−1β by cleaving its precursor form [35]. In this article, we demonstrated the regulatory role of DnaJ in inflammasome-mediated expression of IL−1β. Treatment with DnaJ acts as a priming signal, activating the NF-κB signaling pathway and inducing IL−1β expression, as shown in Figure 2A. Additionally, we have identified ERK as a positive mediator of IL−1β expression alongside NF-κB. Interestingly, DnaJ treatment also serves as the second signal, leading to the activation of caspase-1 and subsequent production and secretion of the mature form of IL−1β (Figure 2B). Furthermore, we discovered that pharmacological inhibition of p38 enhances the level of IL−1β production and GSDMD cleavage by modulating the signaling cascade of the inflammasome. These studies suggest that DnaJ initiates two reciprocal signals, modulating the production of IL−1β by engaging the interplay of p38 and ERK signaling pathways on the inflammasome.

To illustrate the inducible effect of DnaJ, we meticulously purified the DnaJ protein from *E. coli* under endotoxin-free experimental conditions. Since lipopolysaccharides (LPS) can stimulate immune cells, the introduction of this compound during the purification process might have contributed to the induction of IL−1β. To mitigate this possibility, we pretreated DnaJ with Triton X-114, effectively removing most of the residual LPS present in the purified protein. Subsequent analysis using the Limulus amebocyte lysate (LAL) assay indicated that the level of endotoxin contamination was less than 0.05 EU/mg of protein, which was insufficient to induce the expression of IL−1β. MAPKs, a crucial group of serine and threonine kinases, play a pivotal role in transmitting various stimuli by initiating a cascade of protein phosphorylation events that ultimately activate transcription factors [36]. In this study, we observed the negative impact of p38 on the modulation of inflammasome activity in response to DnaJ (Figure 2G). This finding is consistent with previous studies that have demonstrated the involvement of p38 in preventing hyperactivation of the NLRP3 inflammasome, leading to reduced caspase-1 activation and IL−1β production [37]. The active regulation of inflammatory responses is essential as an overactive inflammasome can trigger uncontrolled inflammation and increase susceptibility to septic shock [38]. Therefore, it is interested in understanding the signaling mechanisms by which DnaJ modulates inflammasome activity.

Crucial signal transduction pathways could arise from the cross-talk between TLR4 and the p38 MAPK signaling due to previous studies indicating that DnaJ derived from *S. pneumoniae* activates murine-bone-marrow-derived dendritic cells in a TLR4-dependent manner [24]. TLR4 also serves as a receptor for DnaK, an HSP70 homolog, from *Toxoplasma gondii*, *Francisella tularensis*, and *P. aeruginosa* [39,40,41], suggesting diverse interaction mechanisms underlying TLR4 signaling. Additionally, TLR2 has been demonstrated to participate in the production of IL−8 in response to GroEL, an HSP60 homolog, derived from *H. pylori* in human gastric epithelial cells [42]. However, these TLRs did not contribute to p38 activation (Figure 4C,E), despite their involvement in the partial induction of IL−1β expression (Figure 4A). Notably, we observed that *P. aeruginosa*-derived DnaJ stimulates TLR7 expression through engagement of TLR4-activated AKT/NF-κB and JNK signaling pathways in dTHP-1 cells [27]. These findings imply that the mechanisms governing TLR4 interactions may be more intricate than currently understood. The specific receptors primarily responsible for p38 activation in response to DnaJ treatment in this study remain unclear. Therefore, our future research aims to screen for related PRRs to establish the precise mechanism by which p38 regulates inflammasome activation and subsequent IL−1β production during DnaJ treatment.

NF-κB and MAPK-dependent mechanisms are known to mediate IL−1β gene transcription [43,44]. In line with this, a previous study demonstrated that inhibition of the NF-κB and ERK pathways reduces caspase-1 activation, resulting in decreased IL−1β secretion during *Listeria monocytogenes* infection [45]. The active form of caspase-1 generates IL−1β, which acts as a potential adjuvant, eliciting robust humoral responses and promoting the expansion of antigen-specific T cells. This highlights the significant role of potent inflammasome stimulators in enhancing the efficacy of adjuvants. Notably, examples such as alum adjuvants, which activate the NLRP3 inflammasome [46,47], and cholera toxin, which is extensively studied as a mucosal adjuvant candidate [48], have been identified thus far. Furthermore, administration of IL−1β with heat-killed *L. monocytogenes* enhances resistance to subsequent challenges with live *L. monocytogenes* [49]. However, the extracellular signaling molecules derived from *L. monocytogenes* that modulate inflammasome activation through NF-κB and ERK pathways have remained unidentified.

## 4. Materials and Methods

### 4.1. Reagents

Proteinase K was acquired from Thermo Fisher Scientific (Waltham, MA, USA). BAY11-7082, SP600125, OxPAPC, and CLI-095 were obtained from Invivogen (San Diego, CA, USA). PD98059 and SB203580 were purchased from A.G. Scientific (San Diego, CA, USA).

### 4.2. Cell Culture

All the media mentioned below were supplemented with 10% heat-inactivated fetal bovine serum (FBS; HyClone, Rockford, IL, USA), penicillin (100 units/mL), and streptomycin (0.1 mg/mL). THP-1 (monocytic cell line) cells were cultured in Roswell Park Memorial Institute 1640 (RPMI 1640; HyClone). To induce differentiation of THP-1 cells, treatment with 50 nM of phorbol-12-myristate-13-acetate was carried out for 48 h, and the resulting cells were designated as dTHP-1 cells (macrophage-like cells) in this study. Unless stated otherwise, dTHP-1 cells were exposed to DnaJ protein at a concentration of 1 μg/mL for 4 h. Cell cultures were maintained at 37 °C in a humidified 5% CO_2_ air-jacketed incubator. To incubate cells, we utilized 99% pure CO_2_ provided by Sejong Gas (Sejong, Republic of Korea), and purified water for humidification.

### 4.3. Purification of DnaJ Protein

DnaJ was purified using the Ni-NTA Purification System (Thermo Fisher Scientific). Phase separation treatment with Triton X-114 was used to remove the residual endotoxin. The concentration of the remaining endotoxin was <0.05 EU/mg protein, which was determined using the LAL Chromogenic Endotoxin Quantitation Kit (Pierce Thermo, Rockford, IL, USA). Protein concentrations were determined using the BCA Protein Assay Kit (Pierce Thermo). The purified protein was stored at −80 °C at 500 μg/mL. To obtain a control extract, the same purification protocol was used for *Escherichia coli* BL21 (DE3) cells harboring a pETDuet-1 vector. The control extract was used to evaluate the effect of DnaJ throughout the study.

### 4.4. Real-Time Quantitative PCR (Q-PCR)

Total RNA was extracted using TRIzol^®^ Reagent following the instruction provided by Invitrogen. Q-PCR was performed using SYBR Green PCR Master Mix (KAPA Biosystems, Woburn, MA, USA). The synthesis of cDNA from total RNA was carried out using the ReverTra Ace qPCR RT kit (Toyobo, Japan). The primer sequences used as follows: human IL−1β 5′-AAACAGATGAAGTGCTCCTTCCAG-3′ and 5′-TGGAGAACACCACTTGTTGCTCCA-3′; human CD91 5′-TGTGGCTGTGTTGAAGGATACC-3′ and 5′-TGCTCGTAGGTGTGATGGTAGA-3′; human CD40 5′-GAAACTGGTGAGTGACTGC-3′ and 5′-CACATTGGAGAAGAAGCC-3′; human ERK1 5′-GGAGGACCTGAATTGTATCA-3′ and 5′-CTCCACTGTGATCCGTTTAT-3′; human ERK2 5′-CAGACCTACTGCCAGAGAAC-3′ and 5′-GATATGGTCATTGCTGAGGT-3′. The reactions were carried out using the CFX96 Real-Time PCR System (Bio-Rad, Hercules, CA, USA) with the following thermal conditions: stage 1, 50 °C for 2 min and 95 °C for 10 min; stage 2, 95 °C for 15 s and 60 °C for 1 min. The relative quantities of mRNA were calculated using the comparative CT method and normalized to human GAPDH (5′-CCCTCCAAAATCAAGTGG-3′ and 5′-CCATCCACAGTCTTCTGG-3′) to account for the amount of RNA used in each reaction.

### 4.5. Immunoblotting Analysis

For cell lysis, cells were placed on ice for 20 min in a lysis buffer composed of 20 mM Tris-HCl (pH 7.4), 50 mM NaCl, 50 mM Na pyrophosphate, 30 mM NaF, 5 μM zinc chloride, 2 mM iodoacetic acid, and 1% Triton X-100 in distilled water, supplemented with 1 mM PMSF and 0.1 mM sodium orthovanadate (Sigma-Aldrich, St. Louis, MO, USA). The lysates were then centrifuged at 20,000× *g* for 20 min at 4 °C, and the protein concentration was measured using the BCA Protein Assay Kit (Pierce Thermo). Approximately 30 μg/well of each protein extract was separated on an 8% SDS-PAGE gel and transferred to a 0.45 μm polyvinylidene difluoride membrane. The membranes were blocked with a 5% non-fat dry milk solution at room temperature for 2 h and incubated with the corresponding protein antibodies overnight at 4 °C: p-p38, p38, p-ERK, ERK, p-IKKαβ, IKKβ, TLR2, vimentin, NLRP3, caspase-1 p20 (D7F10), cleaved GSDMD, and β-actin (D6A8) (Cell Signaling Technology, Danvers, MA, USA). The immunoblots were carried out using only one antibody per blot. After washing, the membranes were incubated with HRP-conjugated anti-mouse or anti-rabbit secondary antibodies at room temperature for 2 h. Protein bands were detected by adding the WEST-ZOL^®^plus Chemiluminescent Substrate (Intron, Seongnam-Si, Republic of Korea) and visualized using the Amersham ImageQuant-800 (Cytiva, Marlborough, MA, USA).

### 4.6. Enzyme-Linked Immunosorbent Assay (ELISA)

The amount of IL−1β released into the supernatants was determined using the human IL−1β ELISA kit (R&D System, Minneapolis, MN, USA) following the manufacturer’s instructions.

### 4.7. siRNA Transfection

siERK1 (#sc-29307), siERK2 (#sc-35335), siCD91 (#sc-40101), siCD40 (#sc-29250), siTLR2 (#sc-40256) were obtained from Santa Cruz Biotechnology (Dallas, TX, USA). To transfect dTHP-1 macrophage-like cells, 5 × 10^5^ cells were transfected with the respective targeting siRNA or a negative control pool of oligonucleotides, control siRNA-A (#sc-37007), at concentrations ranging from 10 to 50 nM using Lipofectamine RNAiMax (Invitrogen, Carlsbad, CA, USA), following the manufacturer’s instructions. After 24–48 h of siRNA transfection, the macrophages were cultured in 5% RPMI for an additional 72 h to reach approximately 80% confluence. Quantitative real-time PCR was performed after an additional 24 h. The knockdown efficiency of siRNA was validated by assessing either the mRNA levels using quantitative real-time PCR or the protein levels using immunoblotting analysis.

### 4.8. Statistical Analysis

Statistical analyses were conducted using the Instat package from GraphPad (GraphPad Software (6.0), San Diego, CA, USA). One-way ANOVA followed by Tukey’s post hoc multiple range test was applied. A significance level of *p* < 0.01 was considered statistically significant.

## 5. Conclusions

Considering the crucial role of inflammasome activation and the resulting IL−1β in early immune responses, compounds with potential inflammasome-activating properties hold promise as potent adjuvant candidates. In our study, we demonstrated that DnaJ derived from *P. aeruginosa* contributes to the induction of inflammasome activation through both the NF-κB and ERK signaling pathways in dTHP-1 macrophage-like cells. Furthermore, our findings suggest that CD91/CD40 likely plays a significant role by initiating the ERK MAPK cascades in response to DnaJ. Additionally, the activation of p38 by DnaJ provides a beneficial mechanism to counteract excessive inflammatory responses mediated by inflammasome activation. These results support the notion that DnaJ can serve as a potent stimulator of the inflammasome, effectively promoting the release of biologically active IL−1β from macrophages.

## Figures and Tables

**Figure 1 ijms-24-15957-f001:**
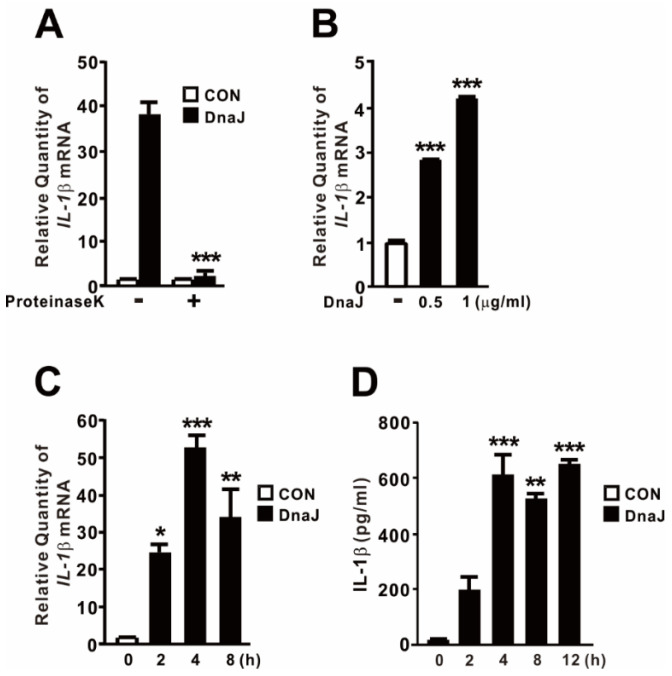
*P. aeruginosa*-derived DnaJ induces the expression of IL−1β. (**A**) Cells were treated for 4 h with recombinant *P. aeruginosa* DnaJ (1 μg/mL), which was pre-treated with proteinase K (20 μg/mL) for 1 h. (**B**–**D**) Cells were treated with either DnaJ at the indicated concentration for 4 h (**B**) or 1 μg/mL of DnaJ for the indicated time (**C**,**D**). After treatment, the increase in IL−1β mRNA level was quantified by qPCR analysis, and the level of IL−1β protein released from cells was measured with supernatant by ELISA analysis. Data are expressed as the mean ± SD (*n* = 3). *, *p* < 0.05; **, *p* < 0.01; ***, *p* < 0.001 vs. DnaJ treatment alone (**A**) or CON. A control extract was used as a negative control (**A**). PBS was used as a control (**B**–**D**).

**Figure 2 ijms-24-15957-f002:**
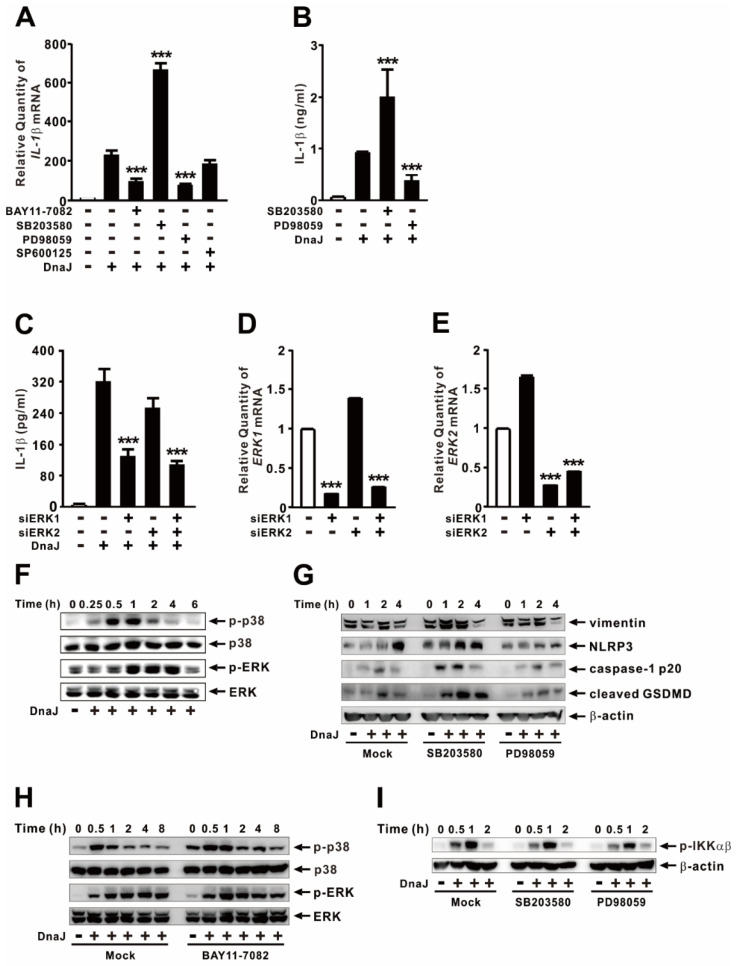
Reciprocal effects of p38 and ERK MAPK signaling pathways for the expression of IL−1β. (**A**,**B**) Cells were pre-treated with either 10 μM of chemical inhibitors (BAY11-7082, SB203580, PD98059, SP600125; (**A**)) or 20 μM of chemical inhibitors (SB203580, PD98059; (**B**)) for 1 h, followed by treatment with 1 μg/mL of DnaJ for 4 h. (**C**–**E**) Cells were transfected with 50 nM of either ERK1 siRNA or ERK2 siRNA. Forty-eight hours post-transfection, the transfected cells were treated with 1 μg/mL of DnaJ for 4 h (**C**). The effect of siRNA was verified by qPCR analysis (**D**,**E**). (**F**) Cells were treated with 1 μg/mL of DnaJ for the indicated time. (**G**–**I**) Cells were pre-treated with 10 μM of chemical inhibitors (SB203580 and PD98059 for (**G**,**I**); BAY11-7082 for (**H**) for 1 h, followed by treatment with 1 μg/mL of DnaJ for the indicated time. After treatment, the increase in IL−1β mRNA level was quantified by qPCR analysis, and the levels of protein were measured by either ELISA or immunoblot analysis. Data in (**A**–**E**) are expressed as means ± SD (*n* = 3). Data in (**F**–**I**) are representative of three separate experiments. ***, *p* < 0.001 vs. DnaJ treatment alone (**A**–**C**), or no transfection (**D**–**E**).

**Figure 3 ijms-24-15957-f003:**
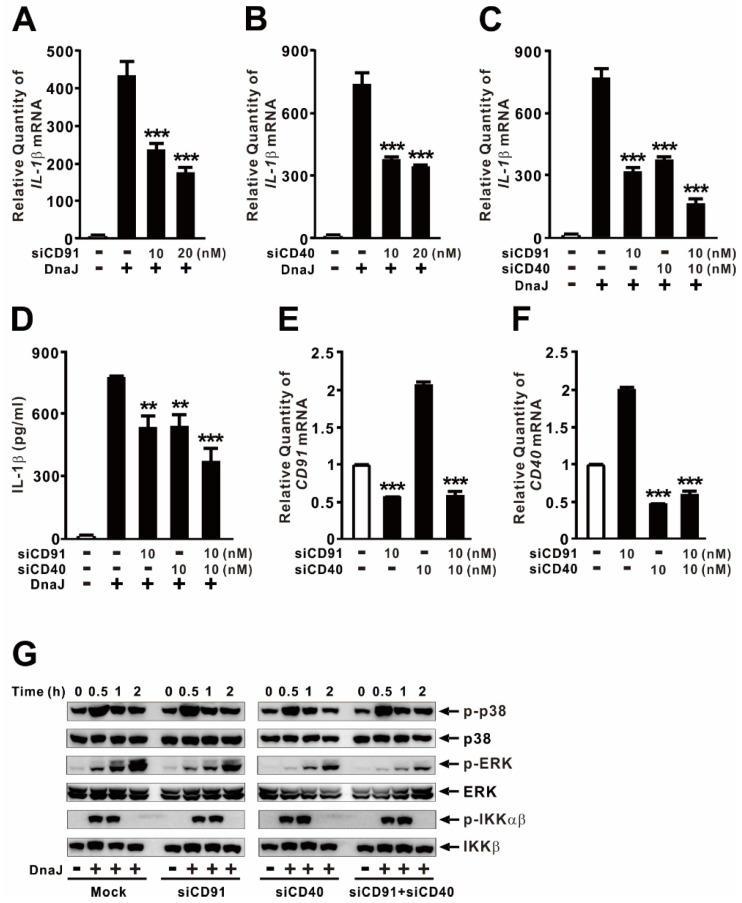
DnaJ-induced IL−1β expression is under the control of CD91/CD40 signaling pathway. Cells were transfected with either 10 or 20 nM of either CD91 siRNA or CD40 siRNA. Forty-eight hours post-transfection, the transfected cells were treated with 1 μg/mL of DnaJ for 4 h (**A**–**D**) or for the indicated time (**G**). The effect of siRNA was verified by qPCR analysis (**E**,**F**). After treatment, the increase in mRNA levels was quantified by qPCR analysis, and the levels of protein were measured by either ELISA or immunoblot analysis. Data in (**A**–**F**) are expressed as means ± SD (*n* = 3). Data in (**G**) are representative of three separate experiments. **, *p* < 0.01; ***, *p* < 0.001 vs. DnaJ treatment alone (**A**–**D**) or no transfection (**E**–**F**).

**Figure 4 ijms-24-15957-f004:**
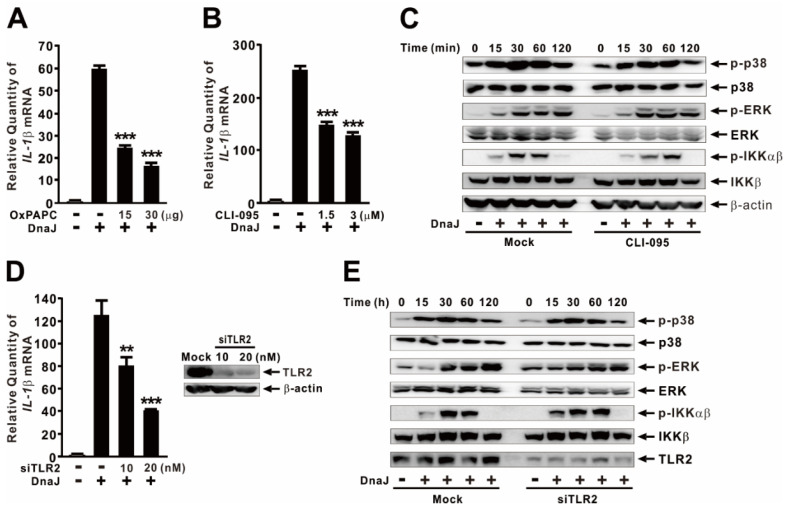
DnaJ-induced IL−1β expression is partially under the control of TLR2/TLR4 signaling pathway. (**A**,**B**) Cells were pre-treated with either OxPAPC (**A**) or CLI-095 (**B**) at the indicated concentrations for 1 h, followed by treatment with 1 μg/mL of DnaJ for 4 h. (**C**) Cells were pre-treated with 3 μM of CLI-095 for 1 h, followed by treatment with 1 μg/mL of DnaJ for the indicated time. (**D**) Cells were transfected with TLR2 siRNA at the indicated concentration. At 24 h post-transfection, the transfected cells were treated with 1 μg/mL of DnaJ for 4 h (*left panel*). The effect of siRNA was verified by immunoblot of TLR2 protein (*right panel*). (**E**) Cells were transfected with 20 nM of TLR2 siRNA. Twenty-four hours post-transfection, the transfected cells were treated with 1 μg/mL of DnaJ for the indicated time. After treatment, the increase in IL−1β mRNA level was quantified by qPCR analysis, and the levels of protein were measured by immunoblot analysis. Data in (**A**,**B**,**D**) are expressed as means ± SD (*n* = 3). Data in (**C**,**E**) are representative of three separate experiments. **, *p* < 0.01; ***, *p* < 0.001 vs. DnaJ treatment alone.

## Data Availability

Not applicable.

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
