# Peer review of "Pseudomonas aeruginosa-Derived DnaJ Induces the Expression of IL−1β by Engaging the Interplay of p38 and ERK Signaling Pathways in Macrophages"

_ijms, 2023, doi:10.3390/ijms242115957_

Round 1
Reviewer 1 Report
Comments and Suggestions for Authors
In the submitted manuscript, Dae-Kyum Kim and collegues, investigate the role of DnaJ from Pseudomonas aeruginosa in macrophage activation, i.e. IL-1β production, by inflammasome triggering.
Overall, the manuscript format is well structured and written with all necessary information in the introduction. However, the materials and methods section needs improvement.
To increase impact of this study and before publication, the authors should revise the manuscript and address concerns raised below.
Concerns:
1. Is the DnaJ preparation from recombinant expression in E.coli “Endotoxin-free”? This should be clarified in the text. It essential to address this question, as LPS contamination impacts on the cell activation experiments.
2. Line 77 ff: For better readability of the manuscript, the authors should include the method of DnaJ preparation in the materials and method section.
3. Line 279: THP-1 cells are a leukemia cell line and thus are used for studies of the human monocyte/macrophage system. These cells do not represent mature and fully differentiated human monocytes and thus need to be termed “monocytic cell line” instead of “human monocytes”, throughout the manuscript. The same issue applies to PMA-differentiated THP-1 cells, which in terms of macrophage markers f.e. CD14, HLA-DR, do not fully recapitulate human primary macrophages. The term “macrophage” for these cells needs to be revised and used more carefully. I would suggest the term “macrophage-like cells” or PMA-THP-1.
Reviewer 2 Report
Comments and Suggestions for Authors
In the manuscript by Kim and co-authors, entitled "Pseudomonas aeruginosa-derived DnaJ induces the expression 2 of IL-1β by engaging the interplay of p38 and ERK signaling 3 pathways in macrophages" the impact of DnaJ on the regulation of IL-1β expression in macrophages. For this purpose the authors examined the effects of DnaJ on the expression of IL-1β, and the roles played by the NF-KB and MAPK signaling pathways on the DnaJ regulation of IL-1β expression at the mRNA level . Results are also presented showing that the IL-1β expression regulation by DnaJ is under the control of CD91/CD40 signaling pathway, as well as unde rpartial control of TLR2/TLR4. The work is well organized and written. There are some issues that the authors should address:
1) The name "Pseudomonas aeruginosa" in the title should be italicized.
2) Immunoblots shown in Figures 2, 3 and 4 are stated as representatives of 3 experiments. The original images in supplementary files only show strips with the chosen part of the immunoblot. In my view, the whole gels should be shown.
3) It is unclear in the experimental part if immunoblots were carried out using only one antiboby per blot or if all th eantibodies of interest were used in the same blot (section 4.4.)
Reviewer 3 Report
Comments and Suggestions for Authors
The authors, Dr. Un-Hwan Ha and colleagues prepared and submitted for review a manuscript entitled “Pseudomonas aeruginosa-derived DnaJ induces the expression of IL-1β by engaging the interplay of p38 and ERK signaling pathways in macrophages”. The authors examined the effect of Dnaj derived from a common soil bacterium Pseudomonas aeruginosa on the regulation of interleukin 1 expression, in particular IL-1β, in macrophages, indicating the importance of understanding the signaling mechanisms underlying IL-1β induction. The discussed research results are important for the reader planning to further deepen knowledge in the field, therefore the manuscript meets the requirements for articles submitted to International Journal of Molecular Sciences and can be published after removing several minor shortcomings and errors, listed below:
· At line 2 is: … … , but should be … … . Comment: So far, the names of microorganisms are written in italics in the title, examples can be found in the titles of current scientific articles: https://doi.org/10.1021/acsabm.3c0064, International Journal of Biological Macromolecules 2023, 253, 127634 (https://doi.org/10.1016/j.ijbiomac.2023.127634), Biomedicine & Pharmacotherapy 2023, 165, 115227 (https://doi.org/10.1016/j.biopha.2023.115227). Please correct. · At line 26 five keywords: DnaJ, ERK, IL-1β, p38, and Pseudomonas aeruginosa are duplicated from the title of the manuscript, therefore they should be removed and several others not included in the title should be suggested, if necessary.
· At line 47 is: … [7-9] … , but should be better … [7–9] … . Comment: Recently, a medium sign “ – “ is used between numbers, not a short one “ - “ as previously used. Please correct throughout the manuscript, in lines 244, 364, 365, 367, 368, 371, 373, 376, 379, 381, 383, 384, 394, 397, 402, 404, 408, 410, 412, 414, 415, 418, 420, 421, 423, 427, 429, 435, 438, 440, 442, 445, 447, 449, 452, and 429. · In line 284, the authors inform about the standard use of CO2, but there is no mention of the purity and supplier of CO2. Please add in the appropriate subsection. Comment: Information about the quality and origin of carbon dioxide as well as water used to humidify the air is important for readers because it may affect the repeatability of experimental results.
· In lines 307 and 326 the authors used mathematical multiplication signs “ ï‚´ “ and “ × “, respectively, but please synchronize their writing style.
· In order to facilitate/enable the study of the manuscript for readers specializing in related fields, the chapter Abbreviations with explanations of the abbreviations used, including the abbreviation Dnaj, is highly recommended.
· Increasing the state of knowledge discussed in the introduction with contemporary source works is extremely desirable, because the authors referred to only 39 literature items, including only four from the last three years, which allows for erroneous opinions about the poor topicality of the research. Comment: Please refer to the current state of knowledge.
